# Combination Therapy with an SGLT2 Inhibitor as Initial Treatment for Type 2 Diabetes: A Systematic Review and Meta-Analysis

**DOI:** 10.3390/jcm8010045

**Published:** 2019-01-04

**Authors:** Tamara Y. Milder, Sophie L. Stocker, Christina Abdel Shaheed, Lucy McGrath-Cadell, Dorit Samocha-Bonet, Jerry R. Greenfield, Richard O. Day

**Affiliations:** 1Department of Diabetes and Endocrinology, St. Vincent’s Hospital, Sydney, NSW 2010, Australia; t.milder@unsw.edu.au (T.Y.M.); j.greenfield@garvan.org.au (J.R.G.); 2Department of Clinical Pharmacology and Toxicology, St. Vincent’s Hospital, Sydney, NSW 2010, Australia; sophie.stocker@svha.org.au; 3St. Vincent’s Clinical School, University of NSW, Sydney, NSW 2010, Australia; lucy.mcgrath-cadell@svha.org.au (L.M.-C.); d.samochabonet@garvan.org.au (D.S.-B.); 4Diabetes and Metabolism Division, Garvan Institute of Medical Research, Sydney, NSW 2010, Australia; 5School of Public Health, Faculty of Medicine and Health, University of Sydney, Sydney, NSW 2006, Australia; christina.abdelshaheed@sydney.edu.au; 6Department of Cardiology, St. Vincent’s Hospital, Sydney, NSW 2010, Australia

**Keywords:** type 2 diabetes, pharmacotherapy, SGLT2 inhibitors, metformin, DPP-4 inhibitors, clinical pharmacology

## Abstract

Background: Guidelines differ with regard to indications for initial combination pharmacotherapy for type 2 diabetes. Aims: To compare the efficacy and safety of (i) sodium-glucose cotransporter 2 (SGLT2) inhibitor combination therapy in treatment-naïve type 2 diabetes adults; (ii) initial high and low dose SGLT2 inhibitor combination therapy. Methods: PubMed, Embase and Cochrane Library were searched for randomised controlled trials (RCTs) of initial SGLT2 combination therapy. Mean difference (MD) for changes from baseline (HbA1c, weight, blood pressure) after 24–26 weeks of treatment and relative risks (RR, safety) were calculated using a random-effects model. Risk of bias and quality of evidence was assessed. Results: In 4 RCTs (*n* = 3749) there was moderate quality evidence that SGLT2 inhibitor/metformin combination therapy resulted in a greater reduction in HbA1c (MD (95% CI); −0.55% (−0.67, −0.43)) and weight (−2.00 kg (−2.34, −1.66)) compared with metformin monotherapy, and a greater reduction in HbA1c (−0.59% (−0.72, −0.46)) and weight (−0.57 kg (−0.89, −0.25)) compared with SGLT2 inhibitor monotherapy. The high dose SGLT2 inhibitor/metformin combination resulted in a similar HbA1c but greater weight reduction; −0.47 kg (−0.88, −0.06) than the low dose combination therapy. The RR of genital infection with combination therapy was 2.22 (95% CI 1.33, 3.72) and 0.69 (95% CI 0.50, 0.96) compared with metformin and SGLT2 inhibitor monotherapy, respectively. The RR of diarrhoea was 2.23 (95% CI 1.46, 3.40) with combination therapy compared with SGLT2 inhibitor monotherapy. Conclusions: Initial SGLT2 inhibitor/metformin combination therapy has glycaemic and weight benefits compared with either agent alone and appears relatively safe. High dose SGLT2 inhibitor/metformin combination therapy appears to have modest weight, but no glycaemic benefits compared with the low dose combination therapy.

## 1. Introduction

Traditionally, a stepwise approach has been used to treat type 2 diabetes, starting with lifestyle interventions and metformin as first-line therapy, and adding second-line therapy if optimal glycaemic control is not achieved. However, an alternative, more intensive approach is initial combination pharmacotherapy. Current guidelines differ in regard to a patient’s glycated haemoglobin (HbA1c) for which initial combination therapy should be considered (American Diabetes Association (ADA) and European Association for the Study of Diabetes (EASD) if HbA1c >1.5% above the patient’s target) or is indicated (American Association of Clinical Endocrinologists (AACE) and the American College of Endocrinology (ACE) if HbA1c >7.5%) [1,2]. Arguments in favour of initial combination therapy rather than metformin monotherapy include avoiding clinical inertia and the potential for a more pronounced and faster improvement in glycaemic control, which may lead to a “legacy effect” [3,4]. In the UK Prospective Diabetes Study (UKPDS), participants newly diagnosed with type 2 diabetes, who were randomised to intensive therapy achieved a lower HbA1c compared with participants randomised to conventional therapy (7.0% versus 7.9%) [5]. After the study ended, this difference in HbA1c between the groups was lost by one year; however after 10 years of follow-up, the intensive therapy group had a reduced risk of myocardial infarction and diabetes-related death compared with the conventional therapy group [6].

Sodium-glucose cotransporter 2 (SGLT2) inhibitors, are potentially an attractive option for initial combination therapy with metformin. These agents have an insulin-independent mode of action—lowering blood glucose concentrations by increasing the urinary excretion of glucose [7]. They also reduce body weight and blood pressure compared with placebo [7]. Empaglilfozin, canagliflozin, and dapagliflozin have been found to significantly reduce the risk of hospitalisation from heart failure among people with type 2 diabetes with established cardiovascular disease or multiple cardiovascular risk factors [8,9,10]. A reduction in risk of a composite outcome of cardiovascular death, myocardial infarction and stroke in people with type 2 diabetes and established atherosclerotic cardiovascular disease with SGLT2 inhibitor use was evident in these cardiovascular outcome trials [11].

A previous meta-analysis compared initial combination therapy of various anti-hyperglycaemic agents and metformin to metformin monotherapy [12]. Only one SGLT2 inhibitor, dapagliflozin, was included in the meta-analysis. Additionally, the efficacy measures in the meta-analysis were restricted to glycaemic control only; effects on body weight and blood pressure were not examined. A recent meta-analysis by Cai et al. [13] examined initial combination therapy in treatment-naïve type 2 diabetes patients, including combination therapy with an SGLT2 inhibitor and metformin. However, not all treatment groups from each trial of combination SGLT2 inhibitor and metformin therapy were included, and there was a discrepancy between the number of subjects included in the efficacy and safety assessments. Furthermore, dose-response relationships were not examined.

Given the limitations of the current literature, we conducted a meta-analysis exploring the efficacy (glycaemic and extra-glycaemic effects) and safety of initial combination therapy with an SGLT2 inhibitor and other anti-hyperglycaemic agents versus monotherapy with each agent in the combination. As two different doses of SGLT2 inhibitors are generally available, we compared the efficacy and safety outcomes of high versus low doses of SGLT2 inhibitor in combination therapy. A greater understanding of the efficacy and safety profile of combination therapy with an SGLT2 inhibitor and other anti-hyperglycaemic agent versus monotherapy with each agent in the combination will aid clinicians when deciding on pharmacotherapy for their newly diagnosed type 2 diabetes patients.

## 2. Methodology

### 2.1. Study Selection

Randomised-controlled trials (RCTs) in treatment-naïve (defined as no pharmacotherapy for at least 12 weeks prior to randomisation) adults with type 2 diabetes were included in the analysis. All dosing regimens of combination therapy that included an SGLT2 inhibitor that were compared to monotherapy (each agent in the combination) were included. The primary outcome was the change in HbA1c from pre-treatment values. Secondary outcomes included the change in body weight, blood pressure (BP) from pre-treatment values, as well as the incidence of adverse events including hypoglycaemia, genital and urinary tract infections (UTIs). All efficacy outcomes were determined after 24-26 weeks of treatment.

### 2.2. Electronic Searches

PubMed, Embase and Cochrane Library were searched (all from inception through to April 2018, without language restrictions) for RCTs based on the following search terms: ‘sodium glucose cotransporter 2’, ‘sodium glucose cotransporter 2 inhibitor*’ and the names of individual SGLT2 inhibitors (full search strategy in Appendix A).

Two independent reviewers (TYM, LMC) screened the titles and abstracts of all the identified articles. The full manuscript of any RCT identified was assessed by two independent reviewers (TYM, LMC) to determine eligibility based on the other selection criteria (as outlined above).

### 2.3. Data Extraction

Two reviewers (TYM, SLS) collected data using a standardised data extraction form. The following data was extracted from the eligible studies: first author, year of publication, study design, sample size, study duration, patient characteristics including mean age, baseline measures (HbA1c, body weight and BP), treatment (type of SGLT2 inhibitor and other anti-hyperglycaemic agent, and the treatment doses), efficacy measures on treatment (HbA1c, body weight and BP) and safety outcome measures.

### 2.4. Risk of Bias Assessment

Two reviewers (TYM/CAS) independently assessed the risk of bias using the Cochrane Collaboration’s tool [14]. Each of the seven items was judged to have a high, low, or unclear risk of bias. A study was considered to have a high risk of bias if one or more domains were judged as being of high risk or if more than half of the domains were considered as unclear risk [14]. Consensus was used to resolve any disagreement.

### 2.5. Measures of Treatment Effect

Data-analysis and generation of figures was carried out using Review Manager version 5.3 (RevMan™, The Nordic Cochrane Centre, The Cochrane Collaboration, Copenhagen, Denmark). Results are presented as mean differences (MD) (95% confidence intervals (CI)) for continuous outcomes such as change in HbA1c, weight and BP. The standard error and mean change score were used to calculate the standard deviation for each of the comparisons. For adverse events outcomes (dichotomous outcomes) the relative risk (RR) and 95% CI was reported.

### 2.6. Assessment of Heterogeneity and Reporting Bias

Clinical heterogeneity was assessed by considering characteristics of the study participants, interventions and outcome measures. Statistical heterogeneity was assessed by visual inspection of forest plots and the *I*^2^ statistic.

### 2.7. Data Synthesis

Studies were grouped based on the outcome measure. A random-effects meta-analysis model was used to combine results where appropriate. In the case of multiple comparisons within the same study, the sample in the control arm was divided by the number of eligible comparisons as per recommendations in the Cochrane handbook [14].

### 2.8. Overall Quality of Evidence Rating

GRADE was used to evaluate the overall quality of evidence [15]. The quality started as high and was downgraded a level for each of the following factors: limitation in study design (when a quarter or more of the studies included in an analysis were considered to have a high risk of bias), inconsistency of results (wide variability in point estimates across studies or if statistical heterogeneity between trials was large (*I*^2^ > 50%)) [16], and imprecision (when the total sample size was <300). Publication bias was not assessed as fewer than ten studies were included in the review. It was also not necessary to downgrade for indirectness as this review specifically evaluated anti-hyperglycaemic treatment for type 2 diabetes. Overall, the quality of evidence was defined as high, moderate, low, and very low [15].

### 2.9. Sub Group and Sensitivity Analysis

A separate analysis was performed for (i) combination SGLT2 inhibitor/metformin therapy versus metformin monotherapy; (ii) combination SGLT2 inhibitor/metformin therapy versus SGLT2 inhibitor monotherapy; (iii) high dose versus low dose SGLT2 inhibitor and metformin combination therapy; (iv) combination SGLT2 inhibitor and non-biguanide anti-hyperglycaemic agent versus SGLT2 inhibitor monotherapy; (v) combination SGLT2 inhibitor and non-biguanide anti-hyperglycaemic agent versus monotherapy with non-biguanide agent.

Importantly, the analysis for the combination therapy versus monotherapy with either an SGLT2 inhibitor or other anti-hyperglycaemic agent used the same dose of SGLT2 inhibitor or other anti-hyperglycaemic agent in combination and monotherapy for comparison.

## 3. Results

A total of 3663 articles were identified initially using the search strategy (Figure 1). Following a review of article titles and abstracts and subsequently inspection of full manuscripts (*n* = 10), four articles (comprising of five studies) were eligible for inclusion in the meta-analysis.

Four studies (*n* = 3749 subjects) from three articles compared initial combination SGLT2 inhibitor and metformin therapy, to either metformin monotherapy or SGLT2 inhibitor monotherapy [17,18,19]. Studies evaluated the combination of metformin and empagliflozin, dapagliflozin or canagliflozin. Participants in these four studies had a mean baseline HbA1c which ranged from 8.7%–9.1% and a mean body weight which ranged from 83–91 kg. One study (*n* = 667 subjects) compared combination therapy with an SGLT2 inhibitor (empagliflozin) and linagliptin, a dipeptidyl peptidase-4 (DPP-4) inhibitor to monotherapy with each agent in the combination [20]. Participants in this study had a mean baseline HbA1c of 8.0% and mean body weight of 88 kg. Details on the study populations, treatments and study durations of the studies included in the meta-analysis are presented in Table 1. The five studies were multiregional.

### 3.1. Risk of Bias Assessments

Risk of bias assessments are included in Table 2. In general, the majority of the domains for the five studies were considered to have a low risk of bias.

### 3.2. Combination SGLT2 Inhibitor and Metformin Therapy versus Metformin Monotherapy

#### 3.2.1. Efficacy

There was moderate quality evidence from the four studies (downgraded for risk of bias) of a statistically significant mean change from baseline (pre-treatment) in HbA1c and body weight at weeks 24–26 favouring the combination therapy (MD (95% CI), HbA1c −0.55% (−0.67, −0.43), weight −2.00 kg (−2.34, −1.66); Figure 2A,B). Two of the four studies reported eligible data on BP (the results from the two studies regarding combination dapagliflozin/metformin therapy [18] were not included as BP in these studies was analysed as a safety measure and included data after rescue medication for hyperglycaemia; whereas analysis of efficacy measures (HbA1c, weight) in these studies did not include data after rescue medication). These two eligible studies showed moderate quality evidence that the combination therapy provides statistically and clinically meaningful reductions in systolic BP and diastolic BP at weeks 24–26 compared with metformin monotherapy (MD (95% CI), −2.35 mmHg (−3.41, −1.30); and −1.37 mmHg (−2.07, −0.67) for systolic and diastolic BP respectively; Figure 2C and Appendix A).

#### 3.2.2. Safety

The four studies showed no statistically significant difference in the incidence of at least one adverse event(s) (AE(s)), drug-related AE(s), serious AE(s), hypoglycaemia, UTI and diarrhoea between the combination therapy and metformin monotherapy (Table 3). However, there was moderate quality evidence of an approximately 2-fold greater risk of genital infection with the combination therapy compared with metformin monotherapy (4.0% vs. 1.8%) with a relative risk (95% CI) of experiencing a genital infection of 2.22 (1.33, 3.72).

### 3.3. Combination SGLT2 Inhibitor and Metformin versus SGLT2 Inhibitor Monotherapy

#### 3.3.1. Efficacy

There was moderate quality evidence from the four studies of a statistically significant mean change from baseline in HbA1c (MD (95% CI), −0.59% (−0.72, −0.46)) and a small but statistically significant reduction in mean body weight (MD (95% CI), −0.57 kg (−0.89, −0.25)) after 24–26 weeks of treatment favouring the combination therapy (Appendix A). The two studies with eligible data on BP provided moderate quality evidence of no difference in systolic BP and diastolic BP from baseline at weeks 24–26 between the combination therapy and SGLT2 inhibitor monotherapy (MD (95% CI), 0.01 mmHg (−0.93, 0.94); and −0.06 mmHg (−0.68, 0.56) for systolic and diastolic BP respectively; Appendix A).

#### 3.3.2. Safety

The four studies showed no statistically significant difference in the incidence of at least one AE(s), drug-related AE(s), serious AE(s), hypoglycaemia or UTI between the combination therapy and SGLT2 inhibitor monotherapy (Table 3). Somewhat surprisingly, there was moderate quality evidence of a lower incidence in genital infections with the combination therapy versus SGLT2 inhibitor alone (4.0% vs. 6.1%) with a relative risk of experiencing a genital infection of 0.69 (95% CI 0.50, 0.96). There was moderate quality evidence of a higher incidence of diarrhoea with the combination therapy compared with SGLT2 inhibitor monotherapy (5.3% vs. 2.3%) with a relative risk of experiencing diarrhoea of 2.23 (95% CI 1.46, 3.40). There was one reported case of diabetic ketoacidosis in the four studies, which occurred in the canagliflozin 300mg group in the study by Rosenstock et al. [21].

### 3.4. High Dose SGLT2 Inhibitor and Metformin Combination Therapy versus Low Dose SGLT2 Inhibitor and Metformin Combination Therapy

#### 3.4.1. Efficacy

The four studies showed moderate quality evidence of no difference in the change from baseline in HbA1c between the high dose and low dose SGLT2 inhibitor and metformin combination therapy at weeks 24–26 (MD (95% CI), 0.02% (−0.08, 0.13)), and a small but statistically significant difference in mean change from baseline in body weight in favour of the higher SGLT2 inhibitor dose; MD (95% CI) −0.47kg (−0.88, −0.06) (Appendix A). Two studies provided moderate quality evidence of no difference in systolic BP and diastolic BP at weeks 24–26 between the high dose and low dose combination therapy (MD (95% CI) −0.04 mmHg (−1.25, 1.17); and 0.25 mmHg (−0.55, 1.06) for systolic and diastolic BP respectively; Appendix A).

#### 3.4.2. Safety

The four studies showed no statistically significant difference in the incidence of safety outcome measures between the high dose and the low dose combination therapy (Table 3).

### 3.5. Combination SGLT2 Inhibitor and DPP-4 Inhibitor versus Monotherapies

#### 3.5.1. Efficacy

There was moderate quality evidence of no difference in the mean change from baseline in HbA1c and body weight between combination empagliflozin/linagliptin and empagliflozin monotherapy after 24 weeks of treatment (MD (95% CI), HbA1c −0.27% (−0.54, 0.00), weight −0.15kg (−0.86, 0.56); Appendix A). This study showed a statistically significant lower mean change from baseline in HbA1c (MD (95% CI), −0.49% (−0.66, −0.32) and body weight (MD (95% CI), −1.55kg (−2.42, −0.69)) with the combination therapy compared with linagliptin monotherapy after 24 weeks of treatment (Appendix A). BP was not analysed as information was only reported on the change from baseline in BP at week 52, not week 24.

#### 3.5.2. Safety

The study showed no statistically significant difference in incidence of safety outcome measures between combination empagliflozin/linagliptin and monotherapy with either empagliflozin or linagliptin alone (Appendix A). However, estimates are limited by the small sample size. No case of diabetic ketoacidosis was reported in this study.

## 4. Discussion

This meta-analysis demonstrates that initial combination therapy with an SGLT2 inhibitor and metformin is more efficacious in terms of glycaemic control and body weight loss than treatment with either drug alone. Further, with the exception of body weight, the glycaemic (HbA1c) and BP responses are maximal with low dose SGLT2 inhibitor in combination with metformin. The frequency of genital infections and diarrhoea is two-fold higher with the combination than with metformin or SGLT2 inhibitor monotherapy, respectively.

Initial combination treatment with an SGLT2 inhibitor and metformin over 24 to 26 weeks results in a clinically meaningful and statistically significant improvement in glycaemic control, weight and BP compared with metformin monotherapy. These results highlight the complementary effects of SGLT2 inhibitors and metformin, which have different mechanisms of actions (SGLT2 inhibitors—glycosuria, natriuresis and caloric loss secondary to inhibition of proximal tubular glucose and sodium reabsorption, and metformin—decrease in hepatic glucose production) [7,22,23,24]. This meta-analysis showed that initial SGLT2 inhibitor and metformin combination therapy lowered HbA1c by a mean of 0.55% compared with metformin monotherapy. This is consistent with a previous meta-analysis that found that initial combination therapy with metformin and another class of anti-hyperglycaemic medication (in half the trials this was a dipeptidyl peptidase-4 (DPP-4) inhibitor) lowered HbA1c by a mean of 0.43% compared with metformin monotherapy over a treatment period of 16 to 76 weeks [12].

Compared with SGLT2 monotherapy, the combination therapy resulted in a greater improvement in glycaemic control and a modest body weight reduction but no difference in BP. This demonstrates that the reductions in BP and to a lesser extent body weight with combination therapy are driven predominantly by the actions of SGLT2 inhibitors, rather than metformin. This is consistent with previous literature [25,26,27].

Although combination therapy resulted in a two-fold increased risk of genital infections compared with metformin monotherapy, there was no difference in the incidence of drug-related adverse events, hypoglycaemia or UTIs. This is consistent with the reported safety profile of SGLT2 inhibitors [8,9]. There was a two-fold increase in the risk of diarrhoea with the combination therapy compared with SGLT2 inhibitor alone. This is consistent with the frequently reported occurrence of gastrointestinal intolerance associated with metformin use [28]. The risk of genital infections was statistically lower with combination therapy compared to SGLT2 inhibitor monotherapy. This is an interesting and unexpected finding and may be related to better glycaemic control in the combination therapy group. In contrast to our results, the meta-analysis by Cai et al. [13] found that combination therapy with an SGLT2 inhibitor and metformin had a higher risk of drug-related adverse events and hypoglycaemia compared with monotherapy with metformin and an SGLT2 inhibitor, respectively. This discrepancy is likely due to Cai et al. not including all study groups from the four studies in their meta-analysis.

Comparison of combination therapy with metformin and high versus low dose SGLT2 inhibitor showed no difference in regard to change from baseline in HbA1c or BP, but there was a modest weight loss benefit with high dose combination (mean difference −0.47 kg). There was no difference in the safety measures between high and low dose SGLT2 inhibitor combination therapies. This suggests the incidences of these adverse events are not related to dose, consistent with previous studies [29,30,31,32]. Baseline characteristics of participants could potentially influence the effect of SGLT2 inhibitor dose on efficacy outcomes, which was not examined.

There was only one RCT that focussed on initial combination therapy with an SGLT2 inhibitor and a DPP-4 inhibitor eligible for inclusion in this systematic review. This study showed no benefit in regard to a reduction in HbA1c or body weight with initial combination empagliflozin/linagliptin, compared with empagliflozin monotherapy. Combination therapy was advantageous in regard to both the lowering of HbA1c and body weight compared with linagliptin monotherapy. Recent meta-analyses by Li et al. [33] and Cho et al. [34] found combination SGLT2 inhibitor and DPP-4 inhibitor therapy resulted in a greater HbA1c reduction compared with DPP-4 or SGLT2 inhibitor monotherapy, respectively. However, the HbA1c difference compared with SGLT2 inhibitor monotherapy was small. In the meta-analyses by Li et al. [33] and Cho et al. [34] combination therapy could be as an add-on to existing diabetes pharmacotherapy, whereas our meta-analysis focussed on combination therapy in treatment-naïve adults.

The strengths of this meta-analysis are first that it only included RCTs and the sample sizes in each treatment group are relatively large. Second, the primary outcome measures of the five RCTs was change from baseline in HbA1c at week 24–26, therefore enabling examination of efficacy measures over a very similar treatment period. Third, efficacy measures were both glycaemic and extra-glycaemic effects. Finally, the dose-response and toxicity relationships of combination therapy with an SGLT2 inhibitor and metformin were explored.

An unavoidable weakness of this meta-analysis was the limited number of eligible studies. However, sample sizes of these studies were considerably large. There were no eligible RCTs regarding combination SGLT2 inhibitor/metformin that included an SGLT2 inhibitor other than empagliflozin, dapagliflozin or canagliflozin. There was only one RCT which examined initial combination therapy with an SGLT2 inhibitor and an anti-hyperglycaemic agent other than metformin, which limits assessment of the efficacy and safety of a SGLT2 inhibitor in combination with a non-biguanide type 2 diabetes drug class. A limitation of our analysis in regard to combination SGLT2 inhibitor/metformin was the inability to pool data on BP from the two studies on combination dapagliflozin/metformin as BP was not an efficacy measure in these studies and was analysed differently from HbA1c and body weight. Additionally, with regard to differences between treatments for certain adverse events, the 95% confidence intervals are broad and whilst no statistical significant difference may have been found, this may be due to a small number of events. Furthermore, these studies did not control for dietary changes of participants, and therefore the impact of diet on efficacy outcomes independent of diabetes pharmacotherapy is unclear.

In summary, our meta-analysis demonstrates that initial combination therapy with an SGLT2 inhibitor and metformin (i) has glycaemic and extra-glycaemic benefits including lowering of body weight and BP and (ii) is relatively safe compared with metformin monotherapy. These extra-glycaemic effects are particularly advantageous given the high incidence of obesity and hypertension among patients with type 2 diabetes, and contrasts with other diabetes drug classes including sulphonylureas and insulin [26,35]. While cardiovascular outcomes were not a focus of the studies included in this meta-analysis, based on the results of cardiovascular outcome trials of empagliflozin, canagliflozin and dapagliflozin, SGLT2 inhibitor and metformin combination therapy potentially also has cardiac benefits among patients with established cardiovascular disease or multiple cardiovascular risk factors [8,9,10,11]. Given the importance of the results of these trials, further research is needed as to whether initial combination therapy with a SGLT2 inhibitor has benefits with regard to cardiovascular outcomes compared with sequential therapy.

Whilst there are clear benefits to initial SGLT2 inhibitor and metformin combination therapy, the cost-effectiveness of this approach is unclear [3,4]. However, the costs of failure to intensify treatment also need to be considered when comparing a stepwise versus combination approach for initial type 2 diabetes treatment, especially given evidence of clinical inertia [36]. In a recent United Kingdom retrospective cohort study, a majority (75%) of type 2 diabetes patients did not receive intensified treatment for more than twelve months after initial indication of monotherapy failure with metformin or a sulphonyurea [37]. Another potential concern with combination therapy is that increasing the number of diabetes medications can be a barrier to adherence [38]. However, single fixed-dose combinations containing an SGLT2 inhibitor and metformin or a DPP-4 inhibitor, respectively are available [39,40].

A key unanswered question is whether the glycaemic and extra-glycaemic benefits of initial SGLT2 inhibitor and metformin combination therapy in patients with type 2 diabetes lead to improved long-term outcomes. Additionally, the cost-effectiveness of this combination approach needs to be examined, as well as an understanding of whether certain patient populations would respond differently from this initial combination therapy. Further research is needed to address these important questions. Finally, this meta-analysis highlights the need for further RCTs examining initial combination with SGLT2 inhibitors and non-biguanide anti-hyperglycaemic agents, especially given that a proportion of patients are either unable to tolerate or have a contraindication(s) to metformin.

## Figures and Tables

**Figure 1 jcm-08-00045-f001:**
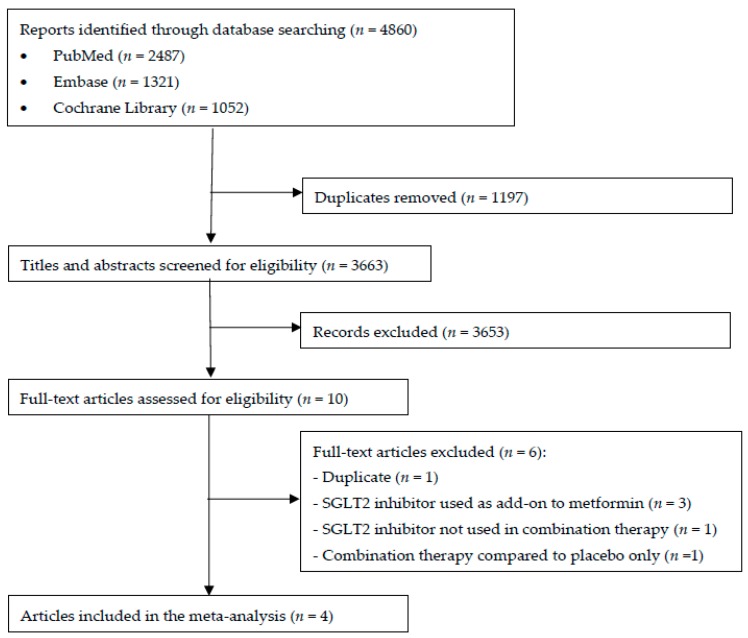
Flow diagram of article inclusion and exclusion.

**Figure 2 jcm-08-00045-f002:**
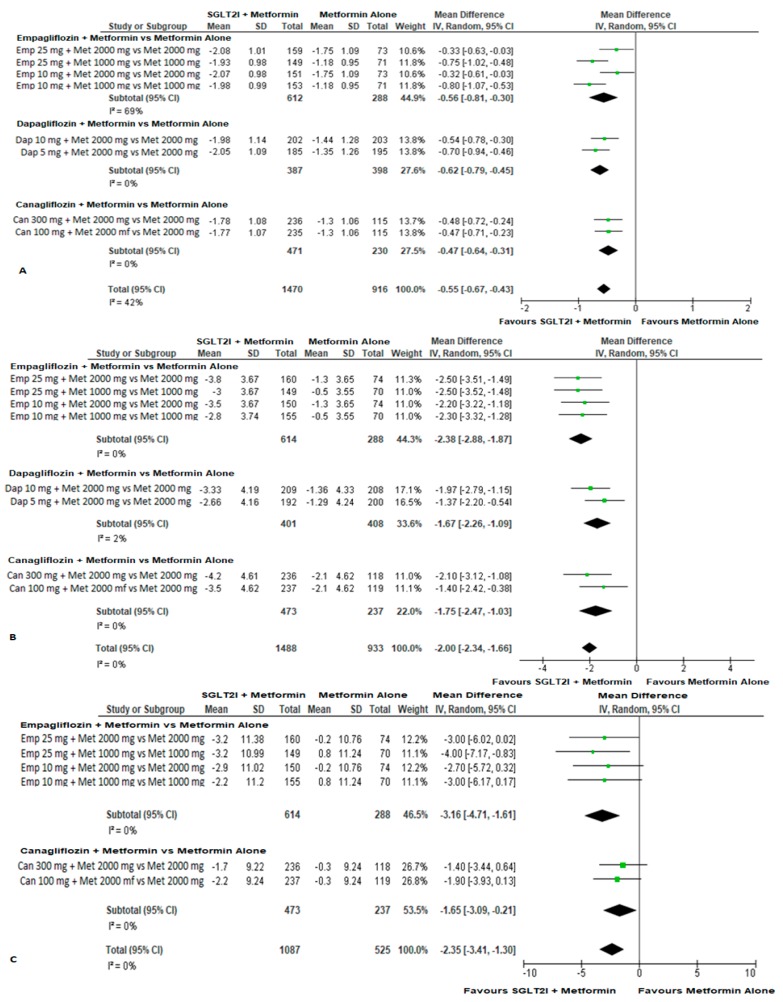
The mean change from baseline (pre-treatment) in: (**A**) HbA1c (%) (**B**) body weight (kg) and (**C**) systolic BP (mmHg) between SGLT2 inhibitor and metformin combination therapy versus metformin monotherapy in treatment naïve type 2 diabetes patients. SGLT2I, SGLT2 inhibitor; Emp, empagliflozin; Met, metformin; Dap, dapagliflozin; Can, canagliflozin.

**Table 1 jcm-08-00045-t001:** Characteristics of the 5 studies including baseline characteristics of participants and details of the interventions.

Study Name, Year, Reference	Study Size, *n*	Mean Age (years)	Sex (%male)	Mean HbA1c (%)	Mean Body Weight (kg)	Combination Therapy Total Daily Doses	SGLT2 Inhibitor Monotherapy Daily Dose	Monotherapy with Other Anti-Hyperglycaemic Agent Total Daily Dose	Study Duration (weeks)
*Combination SGLT2 inhibitor and metformin*									
Hadjadj et al., 2016 [17]	1327	53	56	8.7	83	Empagliflozin 25 mg + metformin 2000 mg Empagliflozin 25 mg + metformin 1000 mg Empagliflozin 10 mg + metformin 2000 mg Empagliflozin 10 mg + metformin 1000 mg	Empagliflozin 25 mg Empagliflozin 10 mg	Metformin 2000 mg Metformin 1000 mg	24
Henry et al., 2012 [18]	638	52	48	9.1	88	Dapagliflozin 10 mg + metformin 2000 mg	Dapagliflozin 10 mg	Metformin 2000 mg	24
Henry et al., 2012 [18]	598	52	44	9.1	85	Dapagliflozin 5 mg + metformin 2000 mg	Dapagliflozin 5 mg	Metformin 2000 mg	24
Rosenstock et al., 2016 [19]	1186	55	48	8.8	91	Canagliflozin 300 mg + metformin 2000 mg Canagliflozin 100 mg + metformin 2000 mg	Canagliflozin 300 mg Canagliflozin 100 mg	Metformin 2000 mg	26
*Combination SGLT2 inhibitor and DPP-4 inhibitor*									
Lewin et al., 2015 [20]	667	55	54	8.0	88	Empagliflozin 25mg + linagliptin 5mg Empagliflozin 10mg + linagliptin 5mg	Empagliflozin 25 mg Empagliflozin 10 mg	Linagliptin 5 mg	52

**Table 2 jcm-08-00045-t002:** The risk of bias assessments for each study using the Cochrane Risk of Bias Tool.

Study Name, Year	Random Sequence Generation	Allocation Concealment	Selective Reporting	Other Bias	Blinding of Participants & Personnel	Blinding of Outcome Assessment	Incomplete Outcome Data	Cochrane Risk of Bias Score
Hadjadj et al., 2016 [17]	Low risk	Low risk	Low risk	Low risk	Low risk	Low risk	High risk *	High risk
Henry et al., 2012 [18]	Low risk	Low risk	Low risk	Low risk	Low risk	Low risk	Low risk	Low risk
Henry et al., 2012 [18]	Low risk	Low risk	Low risk	Low risk	Low risk	Low risk	Low risk	Low risk
Rosenstock et al., 2016 [19]	Low risk	Unclear risk	Low risk	Low risk	Low risk	Low risk	Low risk	Low risk
Lewin et al., 2015 [20]	Low risk	Low risk	Low risk	Low risk	Low risk	Low risk	High risk *	High risk

* Data for an outcome measure (weight in study by Hadjadj et al. [17] and HbA1c in study by Lewin et al. [20]) was collected from fewer than 85% of the total number randomised to one arm of the study.

**Table 3 jcm-08-00045-t003:** Safety outcomes of studies (*n* = 4) of the SGLT2 inhibitor and metformin combination therapy compared with metformin or SGLT2 inhibitor monotherapy in treatment naïve type 2 diabetes patients.

Safety Outcome	Comparator 1	Comparator 2	*I* ^2^	RR (95% CI)
	Number of Events/Total Subjects	Number of Events/Total Subjects		
**i. SGLT2 inhibitor + metformin vs. metformin monotherapy**				
	SGLT2 inhibitor + metformin	Metformin monotherapy		
≥1 AE(s)	886/1559	535/987	0%	1.05 (0.98, 1.13)
≥1 drug-related AE(s)	219/1559	119/987	0%	1.22 (0.98, 1.50)
≥1 serious AE(s)	33/1559	24/987	0%	0.85 (0.49, 1.46)
Hypoglycaemia *	42/1559	19/987	0%	1.20 (0.70, 2.06)
UTI	82/1559	43/987	0%	1.12 (0.77, 1.61)
Events suggestive of genital infection	63/1559	18/987	0%	2.22 (1.33, 3.72)
Diarrhoea	82/1559	67/987	56%	0.86 (0.50, 1.48)
**ii. SGLT2 inhibitor + metformin vs. SGLT2 inhibitor monotherapy**				
	SGLT2 inhibitor + metformin	SGLT2 inhibitor monotherapy		
≥1 AE(s)	886/1559	629/1236	43%	1.08 (0.99, 1.17)
≥1 drug-related AE(s)	219/1559	164/1236	55%	1.06 (0.79, 1.43)
≥1 serious AE(s)	33/1559	29/1236	0%	0.94 (0.56, 1.57)
Hypoglycaemia *	42/1559	20/1236	0%	1.67 (0.99, 2.83)
UTI	82/1559	59/1236	0%	0.97 (0.69, 1.37)
Events suggestive of genital infection	63/1559	76/1236	0%	0.69 (0.50, 0.96)
Diarrhoea	82/1559	29/1236	0%	2.23 (1.46, 3.40)
**iii. High dose SGLT2 inhibitor + metformin vs. low dose SGLT2 inhibitor + metformin**				
	High dose SGLT2 inhibitor + metformin	Low dose SGLT2 inhibitor + metformin		
≥1 AE(s)	445/788	441/771	29%	0.98 (0.89, 1.08)
≥1 drug-related AE(s)	127/788	92/771	64%	1.38 (0.90, 2.12)
≥1 serious AE(s)	15/788	18/771	17%	0.79 (0.36, 1.76)
Hypoglycaemia *	26/788	16/771	0%	1.48 (0.80, 2.75)
UTI	48/788	34/771	34%	1.37 (0.79, 2.39)
Events suggestive of genital infection	38/788	25/771	0%	1.43 (0.87, 2.35)
Diarrhoea	43/788	39/771	15%	1.06 (0.66, 1.69)

RR, relative risk; AE, adverse event; UTI, urinary tract infection * In RCTs by Hadjadj et al. [17] and Rosenstock et al. [19], hypoglycaemia defined as plasma glucose ≤ 3.9 mmol/L and/or assistance required. In RCTs by Henry et al. [18], hypoglycaemia defined as minor (plasma glucose < 3.5 mmol/L) and major (plasma glucose < 3 mmol/L, symptomatic, requiring assistance and prompt recovery after glucose or glucagon administration), or suggestive episode but not meeting criteria for major or minor episode.

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
