# Peer review of "Combination Therapy with an SGLT2 Inhibitor as Initial Treatment for Type 2 Diabetes: A Systematic Review and Meta-Analysis"

_jcm, 2019, doi:10.3390/jcm8010045_

Round 1
Reviewer 1 Report
Milder YT. et al., were collected information from previously published articles on the beneficial effects of Combination therapy with SGLT2 inhibitors and metformin as initial treatment for type 2 diabetes.
Authors analyzed the previously published data on efficacy and safety of initial combination therapy with an SGLT2 inhibitor and other anti-hyperglycaemic agents very thoroughly and well presented.
Design of the study is appropriate enough (ex: exclusion of blood pressure data from two studies (Henry et al.,)) to achieve the aim mentioned in manuscript. Current article has interesting collection of work which, is beneficial especially to clinicians and researchers in diabetes field. Overall study is suggesting the usefulness of Initial combination therapy with SGLT2 inhibitor/metformin with its glycaemic and weight benefits compared to mono therapy with either agent.
I had no specific suggestions for the authors.
Reviewer 2 Report
The authors present a well-conducted meta-analysis on pharmacotherapy strategies for a better management of type 2 diabetes, especially in relation to HbA1c, weight and blood pressure. I find the manuscript well-organised, the rationale sufficiently elaborated, and the methodology appropriate. That being said, I find a hard time acknowledging the value of a meta-analysis that consists in four studies only. I’m aware that the lack of research on this particular topic might be a limitation, but I question whether a meta-analysis is the best choice under such circumstances. Even more when not all studies include adequate placebo groups.
Also, I find interesting that none of the studies analysed control for dietary changes or energy intake, as it is known that hypocaloric diets do have an impact on glycaemic response. I would like to know the opinion of the authors on this, since they don’t mention it either.
Nonetheless, I acknowledge these aspects out of the control of the authors, and based on the content of the manuscript only.
Reviewer 3 Report
In the present study, Tamara Y Milder et al. carried out a systematic review and meta-analysis to study the efficacy and safety of SGLT2 inhibitor/metformin combination therapy among treatment-naïve type 2 diabetes patients. 4 RCTs comprising 3749 subjects were selected. An assessment was performed after 24-26 weeks of treatment. The authors found that SGLT2 inhibitor/metformin combination therapy resulted in a greater reduction in HbA1c and weight compared with either monotherapy. Treatment with SGLT2 inhibitor was associated with a doubling of genital infection and metformin with diarrhea. This is an interesting and important work, which will likely acquire a broad readership and frequent citation.
[Major points]
1) To compare and discuss the effects of high and low doses of a drug, it is important to consider baseline characteristics of the patients enrolled and the optimal speed and extent of improvement in the biomarker/indicator of interest. A too rapid decline in glycemic control may enhance diabetic vitreous hemorrhage. A too rapid body weight loss may indicate dehydration. As to BP, physicians will avoid hypotension. These matters are independent of the robustness and dose-response correlation of SGLT2 inhibitors. For example, even though not significantly, high dose of dapper (mean body weight at enrollment 86.5kg) or cana (91kg) showed a tendency to cause a larger body weight reduction compared to low dose treatment. This tendency was less obvious for empa study (83kg) in which less obese subjects were enrolled. Such carefulness is required to discuss the high/low dose comparison of SGLT2 inhibitor as to HbA1c and BP.
2) Especially, if BP is well controlled before initiation or after starting SGLT2 inhibitors in respective RCTs, physicians will reduce other anti-hypertensive drugs to avoid SGLT2 inhibitor-induced hypotension.
3) Risk of bias is very important in a systematic review. Such information in Table S2 should be presented as a standard Table.
[Minor points]
1) I agree with the authors that a reduced incidence of genital infection with combination therapy compared to SGLT2 inhibitor monotherapy may be caused by a better glycemic control in the combination therapy.
2) Line 29, page 1: (MD(95%CI); >>> (MD[95%CI];
3) Lines 33 and 35, page 1: 95%CI should be described for relative risk. If the word limit for the abstract is a problem, adverse effect of diarrhea may be removed.
